# Arabic Emotional Voice Conversion Using English Pre-Trained StarGANv2-VC-Based Model

Ali H. Meftah [1,*], Yousef A. Alotaibi [1] and Sid-Ahmed Selouani [2]

1 Department of Computer Engineering, College of Computer and Information Sciences, King Saud University, Riyadh 11543, Saudi Arabia
2 LARIHS Laboratory, Campus Shippagan, Université de Moncton, Shippagan, NB E8S 1P6, Canada
* Correspondence: ameftah@ksu.edu.sa

**Abstract:** The goal of emotional voice conversion (EVC) is to convert the emotion of a speaker's voice from one state to another while maintaining the original speaker's identity and the linguistic substance of the message. Research on EVC in the Arabic language is well behind that conducted on languages with a wider distribution, such as English. The primary objective of this study is to determine whether Arabic emotions may be converted using a model trained for another language. In this work, we used an unsupervised many-to-many non-parallel generative adversarial network (GAN) voice conversion (VC) model called StarGANv2-VC to perform an Arabic EVC (A-EVC). The latter is realized by using pre-trained phoneme-level automatic speech recognition (ASR) and fundamental frequency (F0) models in the English language. The generated voice is evaluated by prosody and spectrum conversion in addition to automatic emotion recognition and speaker identification using a convolutional recurrent neural network (CRNN). The results of the evaluation indicated that male voices were scored higher than female voices and that the evaluation score for the conversion from neutral to other emotions was higher than the evaluation scores for the conversion of other emotions.

**Keywords:** emotional voice conversion; voice conversion; emotion; Arabic; GANs; StarGAN

## 1. Introduction

Human speech is a sophisticated signal that contains significantly more information than lexical words. Only 7% of the message is communicated by spoken words or verbal attitude; 38% is communicated by non-verbal vocal characteristics, which reflect emotional state; 55% is transmitted by facial expression [1]. Non-linguistic information, such as emotional state, plays an essential role in human social interaction [2].

In recent years, interaction with machines has become commonplace in various aspects, such as entertainment, security, and healthcare, due to tremendous technological progress [3–5]. One of the most critical aspects of human–machine interaction is voice interaction. Despite developments that have been made, many of the current systems cannot express emotion and the resulting voice does not seem natural because of the lack of essential emotions.

Voice transformation (VT) intends to modify one or more characteristics of a speech signal without altering its linguistic content. It can be used to change a speaker's speaking style, such as switching from neutral to emotional speech or mimicking a native speaker's accent. VT can also be used to change the identification of the speaker by the voice conversion (VC) approach. VC seeks to modify the speech of a source speaker so that the output is viewed as a sentence spoken by a target speaker [6]. Emotional voice conversion (EVC) is a subset of VT, which refers to converting the emotion from the source utterance to the target utterance without changing the linguistic information or speaker identity [7,8].

It is necessary for us to convey our intentions, feelings, and social attitudes through the emotional expression that we use in our everyday talking. Therefore, one of the most

important aspects of speech synthesis is the generation of speech signals with designated emotional attributes [7]. The interest in virtual assistants has grown due to advances in speech processing systems, dialogue systems, and natural language processing. The synthesis of emotional speech by machines bears enormous application potential, particularly in human–machine communication. People would feel more comfortable if the virtual assistant was emotionally controlled, giving it great commercial and social value. Devices such as Alexa, Cortana, and Siri are currently able to broadly synthesize speech, but not yet in different varieties of emotions [8,9]. A human-like speech interaction between humans and machines can be achieved through the ability of EVC to control synthetic speech emotion [8].

EVC's primary challenge is the low availability of training data, especially for under-resourced languages such as Arabic. Indeed, speech collection with emotion annotation necessitates the performance of the required emotions by trained specialists. Another difficulty is that people may disagree with the emotion that is being expressed. It is also challenging to assess emotion control mechanisms due to the subjectivity of speech [7,8]. There is relatively little progress in the field of EVC due to the complexities of emotional features.

Deep neural networks have enabled several EVC frameworks to achieve higher performance than frameworks based on conventional models. Recently, generative adversarial networks (GANs), with their different variants (e.g., cycle-consistent generative adversarial network (CycleGAN, StarGANs, etc.), and autoencoder are the most common techniques used in EVC frameworks [1,10].

There is a significant gap in the research between Arabic and other more widely disseminated languages such as English and Chinese. We might argue that studies on this topic are extremely rare and scarce, where we found only this study [11] on emotional Arabic voice conversion (A-EVC). To the best of our knowledge, this is the first study on an Arabic emotional voice conversion (A-EVC) using StarGANv2-VC.

The lack of Arabic emotional databases, and the scarcity of studies in this field, represent a significant challenge in A-EVC. This work represents the starting point of our project. The primary goal is to answer the following questions: Is the performance of an A-EVC using a pre-trained automatic speech recognition (ASR) network and fundamental frequency (F0) models in another language possible? How good is the quality of the converted voice? To answer these questions, we performed an A-EVC using the StarGANv2-VC model [12] and emotional Arabic speech corpus KSUEmotions [13]. F0 and ASR models are trained in English using the CSTR's VCTK Corpus (Centre for Speech Technology Voice Cloning Toolkit) [14]. The remaining sections of this work are arranged as follows: The second section contains a literature review, while the third section discusses the StarGANv2-VC model. In Sections 4 and 5, respectively, the details of the experiments and the evaluation are described. Section 6 contains the conclusions.

## 2. Literature Review

Earlier studies on voice conversion primarily relied on parallel training data (parallel VC), referring to a pair of utterances with the same content but distinct emotions from the same speaker. Through the paired feature vectors, the conversion model learns a mapping from the source to the target emotion during training [1,6]. Many parallel VC studies adopt a sequence-to-sequence conversion method and encoder–decoder sequence-to-sequence architecture with an attention mechanism. Spectrum [15], prosody features (F0 and energy contour) [15,16], spectral features [17,18], intensity [16], and mel-spectrograms [19] are extracted simultaneously from both the source and the target emotional voice. Deep bidirectional long short-term memory (DBLSTM) [15], artificial neural networks (ANNs), deep belief networks (DBNs) [17], dual supervised adversarial networks [20], recurrent neural networks (RNN), and LSTM are used to propose emotional voice conversion frameworks for German, Telugu, Japanese, and English languages.

Many VC techniques (including those mentioned above) are part of the parallel VC method, which uses training data for the parallel vocalization pairs to learn to map. However, this method has two significant drawbacks: first, obtaining data, which is frequently time-consuming or impractical, and second, using time alignment, which most VC methods rely on but which occasionally fails and necessitates other time-consuming operations, such as pre-checking the exact or manual correction [21].

As a solution to the parallel VC issues, a non-parallel VC that refers to the multi-emotion utterances that do not share the same lexical content across emotions [1,6] has been presented. Zhu et al. [22] suggested a CycleGAN model for unpaired image-to-image translation that learns without paired samples. Many different non-parallel training data for the emotive VC framework are employed with the CycleGAN schema. Zhou et al. [23] performed spectrum and prosody conversion based on CycleGAN, Shankar et al. [24] presented a hybrid architecture that depends on a CycleGAN, and Liu et al. [2] examined how effective the use of CycleGAN was for the Chinese emotional VC task. The fundamental frequency energy contour as well as the mel-frequency cepstral coefficients are the features that are utilized to carry out VC tasks [2,23,24].

StarGAN schema was also applied for EVC. Kameoka et al. [25] used StarGAN, which allows non-parallel many-to-many VC with no parallel utterances, transcriptions, or time-alignment procedures for speech generator training. Rizos et al. [26] used a StarGAN similar to that described in [25] to convert real emotional speech samples into various target emotions as a data augmentation method; it was then verified using a multi-class speech affect recognition test. Furthermore, Moritani et al. [7] employed the same network architecture used in [25]. They investigated how well StarGAN-VC can achieve non-parallel many-to-many EVC for Japanese phrases when applied only to spectral envelope transformation.

Hsu et al. [27] developed a non-parallel VC system with a variational autoencoding Wasserstein GANs (VAW-GAN). Zhou et al. in [28,29] developed a speaker-independent EVC framework based on VAE-GAN [27]. The main aim was to convert anyone's emotion without needing parallel data. Gao et al. [30] suggested an unsupervised non-parallel emotional speech conversion based on style transfer autoencoders. Their model is simple to use in real-world scenarios and does not require paired data, transcripts, or temporal alignment. In [31], Cao et al. used another non-parallel emotional speech conversion method in the English language based on VAE-GAN by improved CycleGAN. Their results show that the submitted method outperforms their baseline [30]. The submitted approach in [32] is closest to the VAW-GAN in [28], where they employed a similar encoder–decoder structure with a VAE encoder with some slight differences, but they operated on mel-spectrograms and trained with multilingual data. In another effort, Shankar et al. [33] extracted contextual pitch and spectral information from parallel data as input features to train a highway neural network by using diffeomorphic curve registration for emotion conversion.

Elgaar et al. [8] utilized the factorized hierarchical variational autoencoder to generate disentangled representations of emotion, hence substantially facilitating emotional VC. With a small amount of emotional speech data, Zhou et al. [34] developed a novel two-stage training technique for sequence-to-sequence emotional VC for both spectrum and prosody. Table 1 provides detailed information on previous studies in this field, illustrating the language, database, selected emotions, the model used, and evaluation metrics in each study.

In summary, there are no significant previous studies related to Arabic speech, as all previous studies were conducted primarily using the English language.

**Table 1.** Details of the previous studies.

| Reference | Year | Language | Selected Emotions | Model |
|---|---|---|---|---|
| Ming et al. [15] | 2016 | English | N, H, F, S | DBLSTM |
| Luo et al. [17] | 2017 | Japanese | N, H, S, A | ANNs and DBNs |
| Luo et al. [20] | 2019 | Japanese | | DSAN |
| Gao et al. [30] | 2019 | English | N, H, S, A | Autoencoders |
| Robinson et al. [35] | 2019 | French | J, F, S, A | Seq2seq |
| Shankar et al. [33] | 2019 | English | N, H, S, A | Highway neural network |
| Rizos et al. [26] | 2020 | English | N, H, S, A | StarGAN |
| Liu et al. [2] | 2020 | Chinese | N, S, A | CycleGAN |
| Liu et al. [36] | 2020 | Chinese | N, S, A | MTEVC |
| Zhou et al. [28] | 2020 | English | N, A | VAW-GAN |
| Zhou et al. [23] | 2020 | English | N, S, A, Sur | CycleGAN |
| Shankar et al. [24] | 2020 | English | N, H, S, A | VCGAN |
| Cao et al. [31] | 2020 | English | N, H, S, A | VAE-GAN |
| Elgaar et al. [8] | 2020 | English | N, H, S, A | FHVAE |
| Vekkot et al. [16] | 2020 | German, Telugu, and English | N, H, A, F | Hybrid network |
| Choi and Hahn [19] | 2021 | Korean | N, H, S, A | Seq2seq |
| Moritani et al. [7] | 2021 | Japanese | N, H, A, J | StarGAN-VC |
| Zhou et al. [34] | 2021 | English | N, H, S, A, Sur | Seq2seq |
| Zhou et al. [29] | 2021 | English | N, A, Sl | VAW-GAN |
| Zhou et al. [37] | 2021 | English and Mandarin | N, H, S, A | VAW-GAN |
| Schnell et al. [32] | 2021 | German and English | Excited and Disappointed | Close to (VAW-GAN) |

N—Neutral, H—Happy, F—Fear, S—Sad, A—Anger, J—Joy, Sur—Surprise, Sl—Sleepy, FHVAE—factorized hierarchical variational autoencoder, DSAN—dual supervised adversarial networks, VCGAN—variational cycle-GAN, Seq2seq—sequence-to-sequence.

## 3. Method

The StarGAN v2 [38] image-to-image translation model creates a mapping between several visual domains with a sufficient amount of variation in the output images and scalability over many domains using a single discriminator and generator. Using the same architecture as StarGAN v2, StarGANv2-VC [12] converts voices by treating each speaker as a separate domain. In our work, we employed the same architecture as StarGANv2-VC [12] for emotional voice conversion and considered each of the following four emotions as a separate domain: neutral, happiness, sadness, and anger. Figure 1 presents an overview of the starGANv2-VC framework, which contains a generator and a discriminator in addition to the F0 network, mapping network, and style encoder.

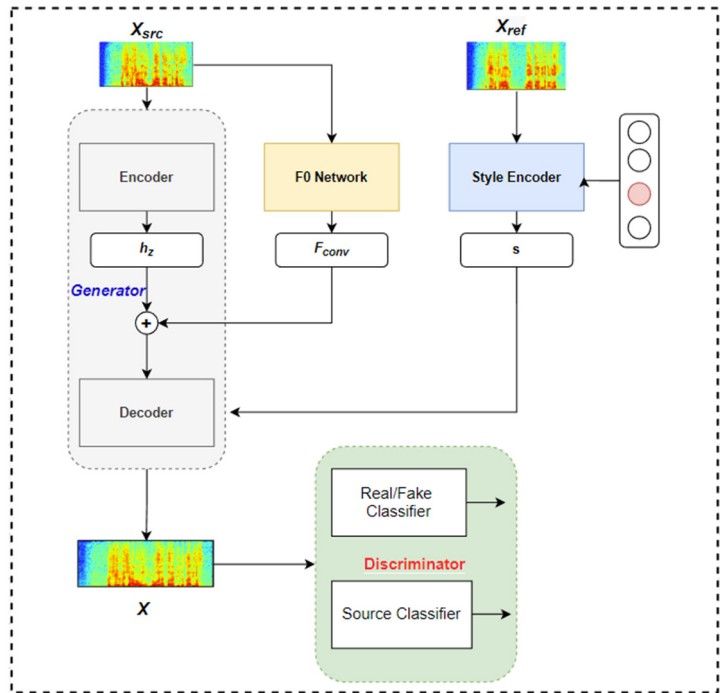

**Figure 1.** StarGAN model framework.

The generator *G* receives an input mel-spectrogram $X_{scr}$ and the fundamental frequency in $h_{f0}$ is converted into output ($X_{scr}$, $h_{str}$, $h_{f0}$) reflective of a domain-specific style code $h_{str}$ either by the style encoder or the mapping network. The primary objective of the **F0 network** is to extract the $h_{f0}$ from a mel-spectrogram input, while the **mapping network** generates a style vector for domain-specific style code and shares for all domains until the last layer is accessed with a random latent code. The latent coding is sampled from a Gaussian distribution to generate diverse style representations across all domains. The **style encoder** extracts the style code for a domain-specific style code and shares it across all domains. **Discriminator *D*** comprises two classifiers that learn the original domain of changed samples. These classifiers share layers that learn the similarities between real and fake samples across all domains to provide feedback regarding features that do not vary from the **generator *G***.

*Training Objectives*

StarGANv2-VC includes adversarial loss ($\mathcal{L}_{adv}$), adversarial source classifier loss ($\mathcal{L}_{advcls}$), style reconstruction loss ($\mathcal{L}_{sty}$), style diversification loss ($\mathcal{L}_{ds}$), F0 consistency loss ($\mathcal{L}_{f0}$), speech consistency loss ($\mathcal{L}_{asr}$), norm consistency loss ($\mathcal{L}_{norm}$), and cycle consistency loss ($\mathcal{L}_{cyc}$). Given a mel-spectrogram $X \in \mathcal{X}_{y_{src}}$, the source and target domains $\mathcal{Y}_{src} \in y$ and $\mathcal{Y}_{trg} \in y$, respectively, in addition to a style code *s*. We trained our model with the following loss functions:

**The generator objective** function calculation is as follows:

$$min_{G,S,M} \; \mathcal{L}_{adv} + \lambda_{advcls}\mathcal{L}_{advcls} + \lambda_{sty}\mathcal{L}_{sty} - \lambda_{ds}\mathcal{L}_{ds} + \lambda_{f0}\mathcal{L}_{f0} + \lambda_{asr}\mathcal{L}_{asr} + \lambda_{norm}\mathcal{L}_{norm} \\ + \lambda_{cyc}\mathcal{L}_{cyc} \tag{1}$$

**The discriminator's objective** function is calculated as follows:

$$min_{C,D} - \mathcal{L}_{adv} + \lambda_{cls}\mathcal{L}_{cls} \tag{2}$$

$\mathcal{L}_{adv}$ denoted to the **adversarial loss** that figures out how to generate a new mel-spectrogram $G(X, s)$ as in Equation (3):

$$\mathcal{L}_{adv} = \mathbb{E}_{X, \mathcal{Y}_{src}}[logD(X, \mathcal{Y}_{src})] + \mathbb{E}_{X, y_{trg}, s}\left[log\big(1 - D(G(X, s), \mathcal{Y}_{trg})\big)\right] \tag{3}$$

where $X$ is the mel-spectrogram that was given as an input and $s$ is a style vector. $X$ and $s$ are the inputs of the generator $G$; $D(., \mathcal{Y})$ is the real or fake classifier's output for the domain $\mathcal{Y} \in y$.

$\mathcal{L}_{advcls}$ denotes the **adversarial source classifier loss** obtained by using the cross-entropy loss function $CE(\cdot)$ with the source classifier as in Equation (4):

$$\mathcal{L}_{advcls} = \mathbb{E}_{X, y_{trg}, s}\left[\boldsymbol{CE}\big(C(G(X, s)), \mathcal{Y}_{trg}\big)\right] \tag{4}$$

$\mathcal{L}_{sty}$ represents the **style reconstruction loss that is applied** to ensure that the generated samples can be used to reconstruct the style code as in Equation (5):

$$\mathcal{L}_{sty} = \mathbb{E}_{X, y_{trg}, s}\left[\| s - S(G(X, s), \mathcal{Y}_{trg}) \|_1\right] \tag{5}$$

$\mathcal{L}_{ds}$ denotes the **style diversification loss** which is used to push the generator to generate a wide variety of samples, each with a unique combination of style codes by maximizing $\mathcal{L}_{ds}$, which is defined in Equation (6):

$$\mathcal{L}_{ds} = \mathbb{E}_{X, s1, s2, y_{trg}}\left[\| G(X, s1) - G(X, s2) \|_1\right] + \mathbb{E}_{X, s1, s2, y_{trg}}\left[\| F_{conv}(G(X, s1)) - F_{conv}(G(X, s2)) \|_1\right] \tag{6}$$

where $s1$ and $s2$ represent two randomly sampled style codes from $\mathcal{Y}_{trg}$; $F_{conv}(.)$ denotes the output of convolutional layers of $F0$ network $F$.

$\mathcal{L}_{f0}$ denotes the **F0 consistency loss** that is applied to obtain F0-consistent results and is defined in Equation (7):

$$\mathcal{L}_{f0} = \mathbb{E}_{X, s}\left[\| \hat{F}(X) - \hat{F}(G(X, s)) \|_1\right] \tag{7}$$

where $\hat{F}(X)$ normalizes the absolute F0 values by its temporal mean.

$\mathcal{L}_{asr}$ denotes **speech consistency loss** that is applied to guarantee that the converted speech contains the same linguistic content as the source. It is defined in Equation (8):

$$\mathcal{L}_{asr} = \mathbb{E}_{X, s}\left[\| h_{asr}(X) - h_{asr}(G(X, s)) \|_1\right] \tag{8}$$

where $h_{asr}(.)$ represents the linguistic feature.

$\mathcal{L}_{norm}$ represents the **norm consistency loss** that is applied for the purpose of preserving the speech/silence intervals of generated samples as in Equation (9):

$$\mathcal{L}_{norm} = \mathbb{E}_{X, s}\left[\frac{1}{T}\sum_{t=1}^{T} \big|\| X., t \| - \| G(X, s)., t \|\big|\right] \tag{9}$$

For a mel-spectrogram $X$ with $N$ mels and $T$ frames at the $t$th frame, the absolute column-sum norm function is performed as follows:

$$\| X., t \| = \sum_{n=1}^{N} | X_{n,t} |$$

where $t \in \{1, \ldots, T\}$ is the frame index.

$\mathcal{L}_{cyc}$ denotes the **cycle consistency loss** that is applied to preserve all other features of the input as defined in Equation (10):

$$\mathcal{L}_{cyc} = \mathbb{E}_{X, \mathcal{Y}_{src}, y_{trg}, s}\left[\| X - G(G(X, s), \check{s})) \|_1\right] \tag{10}$$

where $\check{s}$ represents the estimated style code of the input in the source domain.

$\mathcal{L}_{cls}$ represents the **source classifier loss** and is given in Equation (11):

$$\mathcal{L}_{cls} = \mathbb{E}_{X,\,y_{src},S}\big[\boldsymbol{CE}(C(G(\boldsymbol{X},s))),y_{src}\big] \tag{11}$$

where $\lambda_{advcls}$, $\lambda_{asr}$, $\lambda_{sty}$, $\lambda_{f0}$, $\lambda_{ds}$, $\lambda_{cyc}$, $\lambda_{norm}$, and $\lambda_{cls}$ are hyperparameters for each term.

## 4. Experiments

### 4.1. Databases

KSUEmotions corpus [13] is a read and acted emotion dataset recorded in two phases for modern standard Arabic (MSA) and published by the Linguistic Data Consortium. In the second phase, each sentence was spoken with five emotions (neutral, sadness, happiness, surprise, and anger) during two separate trials. This phase includes seven male speakers, seven females, ten sentences of medium length, and two sentences consisting of only one word, either yes or no. The sampling frequency is 16,000 Hz and the total duration of the recording is two hours and 21 min in Phase 2. Table 2 shows the details of the KSUEmotions Phase 2 database.

**Table 2.** KSUEmotions Phase 2 details.

| | |
|---|---|
| **Sentences** | S05–S12, S15, S16 |
| **Speakers** | 7 males, 7 females |
| **Average Sentence Duration** | 4 s |
| **Corpus Duration** | 141 min |
| **Total Audio files** | 14 (Speakers) × 12 (Sentences) × 2 (Trials) × 5 (Emotions) = 1680 files |

### 4.2. Data Preparation

The experiments for this work were only performed on the KSUEmotions Phase 2 database. According to the relevant literature, and as shown in Table 3, the four emotions most frequently used in EVC are neutral, happiness, sadness, and anger. As a result, we limited our experiments to these four emotions. The data were divided into 81% for training and evaluation and 19% for testing in a way that came very near to being fully random. Both the training dataset and the test dataset were partitioned in a way that prevented any particular sentence from the test dataset from appearing in the training dataset for the same speaker, as shown in Table 3.

**Table 3.** Data preparation (selection of the sentence and speaker for the test sets in each emotion) (N—neutral, H—happiness, S—sadness, A—anger; 20 represents the total number the selected files).

| Source | | N | | | | H | | | | S | | | | A | | | Total |
|---|---|---|---|---|---|---|---|---|---|---|---|---|---|---|---|---|---|
| **Target** | **N** | **H** | **S** | **A** | **N** | **H** | **S** | **A** | **N** | **H** | **S** | **A** | **N** | **H** | **S** | **A** | |
| | | Sent.5/Speaker 1 | | | | Sent.6/Speaker 2 | | | | Sent.7/Speaker 3 | | | | Sent.8/Speaker 4 | | | 16 |
| | | Sent.9/Speaker 5 | | | | Sent.10/Speaker 6 | | | | Sent.11/Speaker 7 | | | | Sent.12/Speaker 1 | | | 16 |
| | | Sent.15/Speaker 2 | | | | Sent.16/Speaker 3 | | | | Sent.5/Speaker 4 | | | | Sent.6/Speaker 5 | | | 16 |
| | | Sent.7/Speaker 6 | | | | Sent.8/Speaker 7 | | | | Sent.9/Speaker 1 | | | | Sent.10/Speaker 2 | | | 16 |
| | | Sent.11/Speaker 3 | | | | Sent.12/Speaker 4 | | | | Sent.15/Speaker 5 | | | | Sent.16/Speaker 6 | | | 16 |
| **Total** | **5 Sent. × 4 emotions = 20** | | | | **20** | | | | **20** | | | | **20** | | | | **80** |

For illustration, we chose sentence S05 to convert from the source (neutral) emotion to the target emotions (neutral, happiness, sadness, and anger). Moreover, we picked sentence S06 to convert the source emotion of happiness to neutral, happiness, sadness, and anger, and so on for the other emotions and sentences as listed in Table 3. The speakers were selected by using the same procedure; we started with the first speaker (male or female) and selected the sentence S05 that he or she uttered. Then, we moved on to the second speaker and selected the sentence S06, and so on until speaker (male or female) number seven, after which the cycle was repeated. The total number of selected files is 320 (80 for

males and 80 for females in two trials, where each trial contains 160 files; one trial was used as a testing set, and the other was dropped).

### 4.3. Implementation Details

We used a mel-spectrogram of our data as an input to the model as shown in Figure 1 and as explained in the training objective section. As we mentioned above, the F0 and ASR models are trained in the English language using the CSTR's VCTK Corpus. Using the ASR and the F0 pretrained model and the AdamW [39] optimizer, we trained the StraGANv2-Vc model for 2000 epochs by setting the parameters as shown in Table 4. The source classifier joined the training process after 50 epochs. Training the F0 model was done with pitch contours obtained from the WORLD vocoder [40].

**Table 4.** Setting of model parameters.

|  |  |  |
|---|---|---|
| StarGANv2-VC Model | No. of Epochs | 2000 |
|  | Learning rate | 0.0001 |
|  | $\lambda_{clc}$ | 0.1 |
|  | $\lambda_{advcls}$ | 0.5 |
|  | $\lambda_{sty}$, $\lambda_{ds}$, $\lambda_{norm}$, $\lambda_{cyc}$, and $\lambda_{asr}$, | 1 |
|  | $\lambda_{f0}$ | 5 |
| F0 Model | No. of Epochs | 100 |
| ASR model (phoneme level) | No. of Epochs | 80 |
|  | character error rate (CER) | 8.53% |

### 4.4. Results

We converted each file for any speaker in the testing set from one emotion to another targeted emotion in addition to the source's emotion. As indicated in Table 3, for each emotion, we selected 20 speech files from male speakers and 20 from female speakers. These 20 speech files were converted into the four emotions, resulting in 80 files for males and 80 files for females. In the end, the total number of converted files was equal to 640 files (we used only one trial, which contains 160 files), as shown in Figure 2.

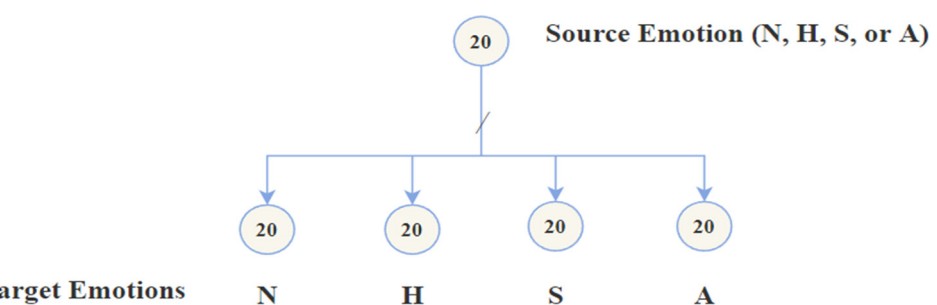

**Figure 2.** Generated files for each emotion (N—neutral, H—happiness, S—sadness, and A—anger).

To illustrate, we selected one audio file (D05E0XP005S11T02) containing sentence S11 displayed below in Arabic and the International Phonetic Alphabet, which was spoken by speaker number 5 with four distinct emotions (N—neutral, H—happiness, S—sadness, and A—anger)

*S11*:　　　　　　　　　السادات بطل الحرب والسلام

　　　　　　　　　*?assaadaat bat?alul ħarbi wassalaam*

Figure 3 presents a spectrogram for the source emotion (S) (first column) and generated files (from second to fifth columns). The first column contains the source emotion of the neutral in the first row, the source emotion of happiness in the second row, and the sadness and anger emotions in the third and fourth rows, respectively. The second column represents

the conversion from source to neutral (X2N), the third column represents the conversion from source to happiness (X2H), followed by the conversion from source to sadness (X2S), and the conversion from source to anger (X2N) in the fourth and fifth columns, respectively, where X is any source emotion (N, H, S, or A). For instance, the second row/first column depicts the spectrogram in its original state (happiness) and the second column displays a spectrogram labeled H2N for converting happiness to neutral. The third column displays a spectrogram labeled H2H for converting happiness to happiness, etc.

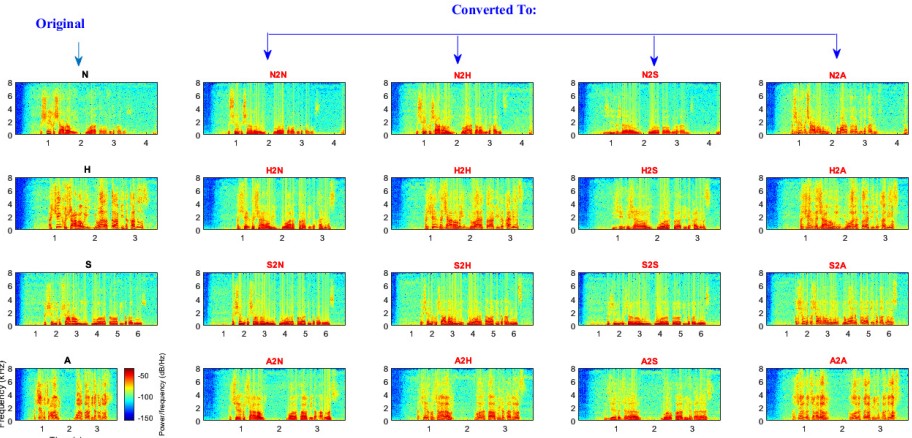

**Figure 3.** Spectrogram of the original (first column) and generated audio file (second to fifth columns) (N2S—source emotion N converted to the target emotion S, etc.).

## 5. Evaluation

The assessment of speech quality produced by conversion and by measuring the system's accuracy compared with state-of-the-art approaches is a fundamental stage in speech conversion. Speaker similarity, speech quality, and speech intelligibility are criteria for evaluating the performance of the VC system. In general, the results are measured in terms of objective and subjective measurements.

The standard objective evaluation measures comprise mel-frequency cepstral distortion (MCD) for spectrum and root-mean-square error (RMSE) for prosody [41]. In the subjective evaluation, which is usually performed by listeners, the lowest number of listeners in the relevant literature review was 10 [17,18,20] and, in some cases, it reached 87 [35]. Collecting a sufficient number of listeners and then preparing them for a human perceptual test to evaluate the generated files is costly in time and money. Due to the significant effort and resources required for subjective evaluation, we decided to perform the evaluation process objectively, as shown in Figure 4.

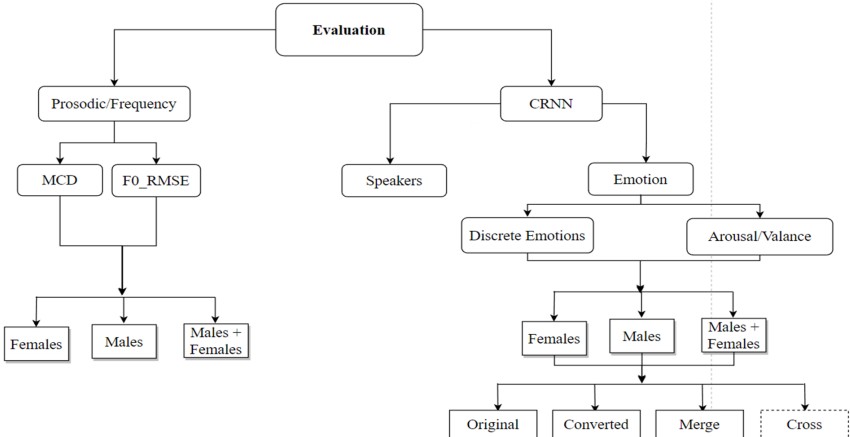

**Figure 4.** Framework of evaluation experiments.

### 5.1. Spectrum and Prosody Conversion

We used MCD, to evaluate the performance of spectrum conversion. MCD compares the converted mel-frequency cepstra to the target mel-frequency cepstra and measures the amount of distortion. In this work, 24-dimensional mel-cepstral coefficients (MCEPs) are extracted from each frame; MCD is defined as [29,42].

$$\text{MCD} = (10/ln10)\sqrt{2\sum_{i=1}^{24}\left(mc_i^t - mc_i^c\right)^2}$$

where the terms $mc_i^t$ and $mc_i^c$ stand for the target and converted mel-frequency cepstra, respectively.

We used root-mean-square error (RMSE) [43], which is a measure for prosody conversion performance and is defined as

$$\text{RMSE} = \sqrt{\frac{1}{N}\sum_{i=1}^{24}\left(\left(F0_i^t\right) - \left(F0_i^c\right)\right)^2}$$

where the terms $F0_i^t$ and $F0_i^c$ stand for the target and converted $F0$ features, respectively. If the MCD and F0-RMSE values are low, this indicates that the distortion or prediction error is reduced as well.

The MCD and F0-RMSE average results of converting one emotion to another for different combinations are reported in Table 5 for the female set, male set, and male and female set (MF).

**Table 5.** The average MCD and F0-RMSE results (low value is better). (MF—Females + Males; overall mean: the average of conversion from source to all other emotion).

|  | MCD (dB) | | | F0_RMSE (Hz) | | |
|---|---|---|---|---|---|---|
|  | **Females** | **Males** | **MF** | **Female** | **Male** | **MF** |
| **N2N** | 5.5 | 5.39 | 5.45 | 41.27 | 27.17 | 27.17 |
| **N2H** | 5.76 | 5.51 | 5.64 | 47.98 | 32.54 | 32.54 |
| **N2S** | 5.55 | 5.55 | 5.55 | 43.3 | 26.12 | 54.65 |
| **N2A** | 5.72 | 5.65 | 5.69 | 43.73 | 27.08 | 62.05 |
| **Overall mean** | 5.63 | 5.53 | 5.58 | 44.07 | 28.23 | 44.10 |
| **H2N** | 6.41 | 6.48 | 6.44 | 47.03 | 73.68 | 61.94 |
| **H2H** | 6.38 | 6.22 | 6.3 | 46.54 | 51.73 | 51.73 |
| **H2S** | 6.58 | 6.51 | 6.55 | 43.57 | 40.27 | 40.27 |
| **H2A** | 6.46 | 6.4 | 6.43 | 41.76 | 75.53 | 69.22 |
| **Overall mean** | 6.46 | 6.40 | 6.43 | 44.73 | 60.30 | 55.79 |
| **S2N** | 5.82 | 5.68 | 5.75 | 56.4 | 31.57 | 35.74 |
| **S2H** | 6.12 | 5.66 | 5.88 | 56.92 | 23.35 | 23.35 |
| **S2S** | 5.64 | 5.38 | 5.5 | 52.34 | 22.22 | 22.22 |
| **S2A** | 6.08 | 5.94 | 6.01 | 57.64 | 53.12 | 53.12 |
| **Overall mean** | 5.92 | 5.67 | 5.79 | 55.83 | 32.57 | 33.61 |
| **A2N** | 6.58 | 6.69 | 6.63 | 78.7 | 50.18 | 50.18 |
| **A2H** | 6.48 | 6.32 | 6.4 | 73.6 | 39.51 | 68.82 |
| **A2S** | 6.66 | 6.88 | 6.76 | 88.55 | 39.24 | 39.24 |
| **A2A** | 6.33 | 6.18 | 6.26 | 79.92 | 67.7 | 67.7 |
| **Overall mean** | 6.51 | 6.52 | 6.51 | 80.19 | 49.16 | 56.49 |

N2A—neutral to anger, etc.

Table 5's left side displays the MCD results, while the right side displays the F0-RMSE results. For more clarity, Figures 5 and 6 show MCD and F0_RMSE for the conversion from one emotion to the other targeted emotions for females, males, and MF. Table 5 and Figures 5 and 6 show that, with a simple exception, males have the lowest values of MCD and F0 RMSE. This conclusion is consistent with our previous study in emotion recognition regarding evaluating the original database, human perceptual test results [44], convolutional recurrent neural network, and residual neural network [45], which indicated that males were evaluated more favorably than females. This result is compatible with our new findings.

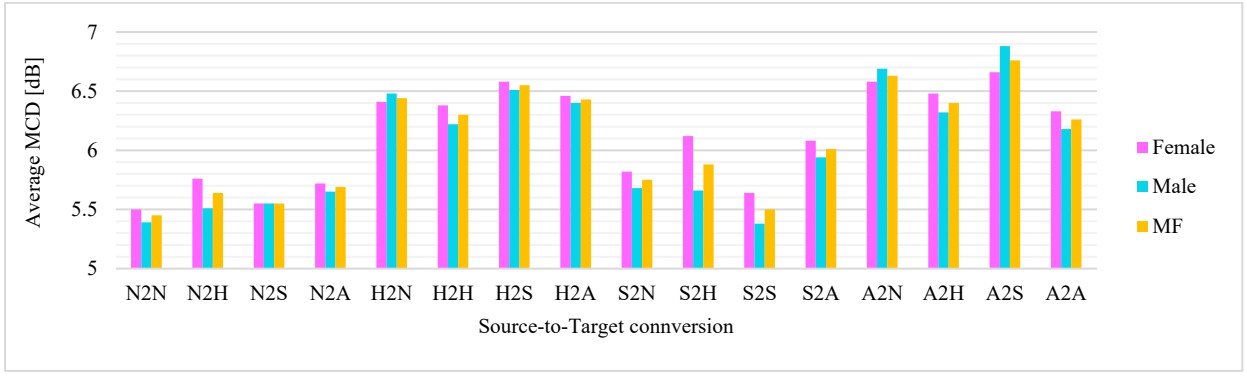

**Figure 5.** Average MCD value for the source-target emotion conversions.

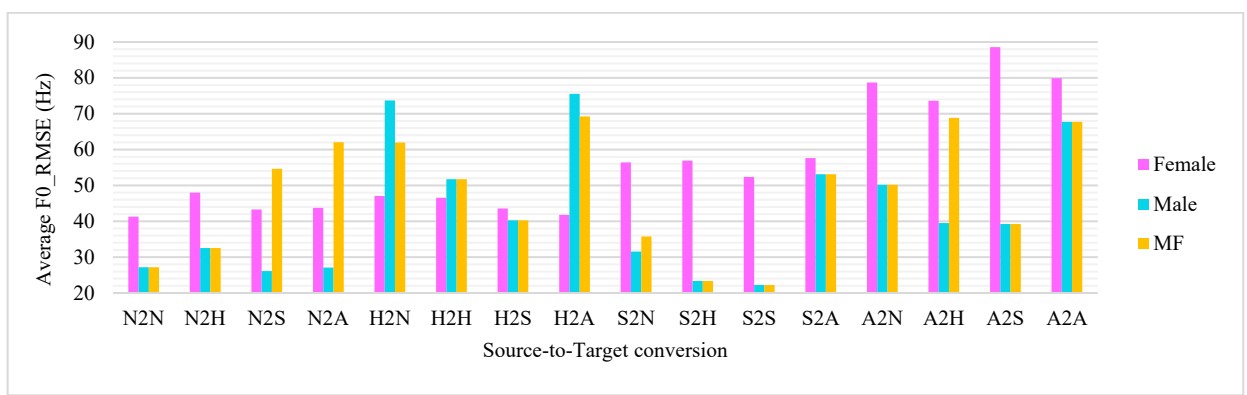

**Figure 6.** Average F0_RMSE value for the source-target emotion conversions.

Figures 7 and 8 show the overall mean of MCD and F0_RMSE for each emotion. As shown in Figure 7, the lowest value of the MCD was noticed when converting from the emotion of neutral to the other different emotions and then when converting from the emotion of sadness to the other emotions. On the other hand, the highest value of MCD was observed while converting from anger to other emotions, followed by a minor difference when converting from happy to other emotions.

As in the MCD result, the value of F0_RMSE is lowest when converting from neutral to other different emotions for males only or females only and when converting from sadness to other emotions for MF. However, the highest value of F0_RMSE was observed during the conversion from anger to other different emotions for females only and MF, and when converting from happiness to other emotions for males.

The best and worst emotion conversion based on MCD and F0 RMSE values regarding gender are presented in Table 6. The latter is based on the findings presented in Table 5 and Figure 5, as well as Figure 6, and compares emotions that have converted from better (which got the lowest value) to worse. Neutral, sadness, happiness, and anger is the order from the best to the worst for females, males, and MF. For F0_RMSE, the best and worst are not in the same style as in MCD, as shown in Table 6, where neutral is the best for males

only or females only and the worst is anger for females and happiness for males. For the MF, sadness, neutral, anger, and happiness are ordered from best to worst.

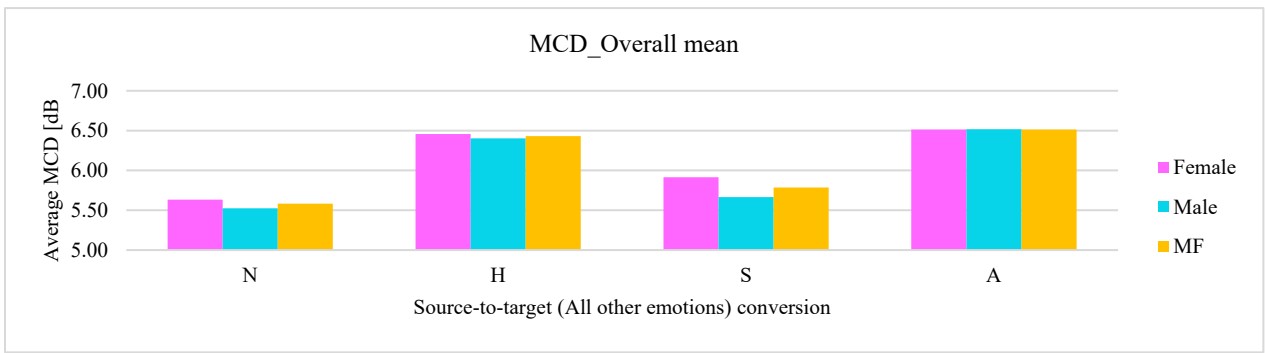

**Figure 7.** Overall mean MCD value for the conversion from source to all other emotions.

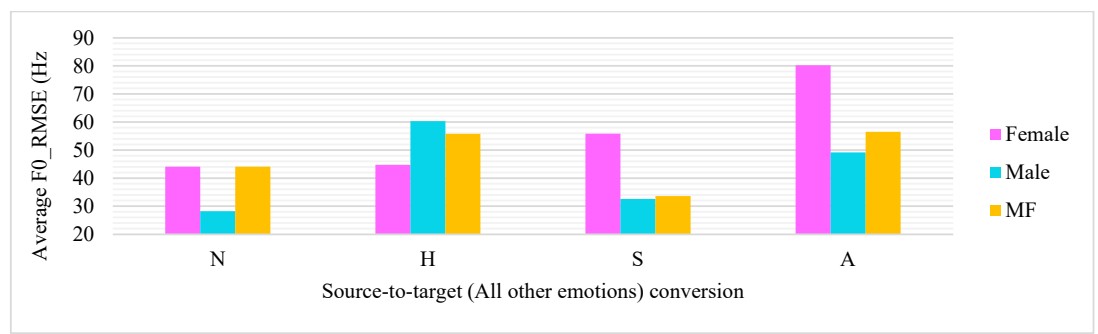

**Figure 8.** Overall F0_RMSE value for the conversion from source to all other emotions.

**Table 6.** Best and worst conversion regarding gender.

| | | From Best (Left) to Worst (Right) | | | |
|---|---|---|---|---|---|
| **Female** | MCD | N | S | H | A |
| | F0_RMSE | N | H | S | A |
| **Male** | MCD | N | S | H | A |
| | F0_RMSE | N | S | A | H |
| **MF** | MCD | N | S | H | A |
| | F0_RMSE | S | N | A | H |

Considering the best and worst emotion conversion from a specific emotion based on MCD and F0 RMSE, each emotion was converted into four emotions (the other three emotions and the emotion itself). The conversion is ranked from best to worst for each emotion in Table 7. As shown in the table, the original emotion is on the left (column called "From Emotion"), followed by the best emotion which converted to it, for females, males, and MF, and the worst for the same categories.

Regarding the MCD values in Table 7, according to what was predicted, the optimum outcome of the conversion was the conversion to the emotion itself; hence, the best outcome for the neutral emotion was the conversion to the neutral emotion, and so on for the rest of the emotions. On the other hand, the worst consequence of converting from the emotion of neutral was the conversion from it to the emotions of anger and happiness. For emotions of happiness and anger, the conversion's worst outcome was when converting to the emotion of sadness for both. For sadness, it was the worst result of its conversion into an emotion of anger for males only, as well as for MF, and when converting into happiness for females.

**Table 7.** Best and worst conversion regarding emotions.

| | | Converted to | | | | | |
| --- | --- | --- | --- | --- | --- | --- | --- |
| From Emotion | | Best | | | Worst | | |
| | | Female | Male | MF | Female | Male | MF |
| MCD | **N** | N | N | N | H | A | A |
| | **H** | H | H | H | S | S | S |
| | **S** | S | S | S | H | A | A |
| | **A** | A | A | A | S | S | S |
| F0_RMSE | **N** | N | S | N | H | H | A |
| | **H** | A | S | S | N | A | A |
| | **S** | S | S | S | A | A | A |
| | **A** | H | S | S | S | A | H |

Regarding the F0_RMSE values, the results for neutral and sadness are, with exceptions, comparable to their findings regarding MCD. When it comes to the emotions of anger and happiness, the data presented in the table reveal that the most favorable outcome for anger occurred when the emotion was converted from anger to happiness, while the most favorable outcome for happiness occurred when anger was converted from happiness to anger in relation to happiness, as shown in Table 7.

*5.2. Automatic Recognition*

5.2.1. Emotion Recognition Experiments

We have conducted several experiments to obtain the best automatic evaluation; these were divided and carried out according to the type of emotion, gender, and data, as shown in Figure 4.

Based on the type of emotion, we defined two primary experiments: the first relates to discrete emotions (neutral, happiness, sadness, and anger), and the second relates to arousal (high and low) and valence (positive and negative) emotions, as shown in Table 8 [46].

**Table 8.** Mapping emotions for valence and arousal discrimination.

| Valence | | Arousal | |
| --- | --- | --- | --- |
| Positive | Negative | Low | High |
| N, H | S, A | N, S | H, A |

Based on gender, the data were divided into three independent sets (females only, males only, and females and males together) and experiments were carried out on each set separately.

The data were divided into four categories: original, converted, merged, and crossed. "Original data" refers to the original KSUEmotions database, while "converted data" refers to the data that were newly generated. "Merged data" includes both the original and the converted data, and in the "crossed data", we used the original data for training and the converted data for testing. The recognition experiments were conducted individually on each type.

5.2.2. Speaker Identification Experiments

To evaluate speaker identification, we conducted several experiments regarding the speaker's gender and data category. The experiments were conducted on a separate set of females, consisting of seven speakers. Next, they were conducted on a set of males, also composed of seven speakers. Finally, they were conducted on all speakers, both male and

female together. All these experiments were carried out in the original database and with converted data, merged data, and crossed data, as described in the previous section.

### 5.2.3. Used Features and Model

As a feature for emotion recognition and speaker identification, the spectrogram was generated for each original and converted audio file. We utilized the CRNN model, which is a combination of a convolutional neural network (CNN) and long short-term memory (LSTM), as demonstrated in Figure 9.

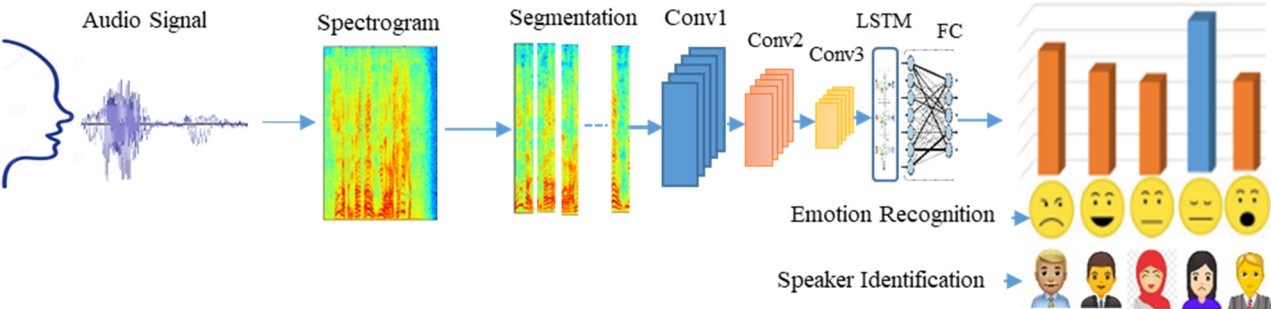

**Figure 9.** Proposed CRNN model.

The spectrogram was segmented after being generated for each audio file. These segments, with a size of $S_s \times 512$, were used as an input for the proposed model, where $S_s$ represents the segment size in frames and 512 is the number of frequency points output by the short-time FFT.

The proposed model contains three convolutional layers: the first one includes 16 kernels of dimension size $12 \times 16$, the second one includes 24 kernels of dimension size $8 \times 12$, and the third one consists of 32 kernels of dimension size $5 \times 7$. These convolutional layers were applied with a one-pixel stride. Each layer is followed by an exponential linear unit (ELU) to help the CNNs train much faster. The ELU in each layer is attached by a max-pooling layer ($2 \times 2$) with a stride of two. Bidirectional-LSTM (BLSTM) layers follow these layers with 128 units in addition to a dropout layer with a dropout ratio of 40% to prevent overfitting. Finally, a fully connected layer (FC) follows with several neurons equal to the number of emotions or speakers. At the end of the FC layer, a sigmoid function was used to make classification outputs in the form of prediction accuracy for different emotions or speakers.

We used a probabilistic evaluation for system prediction using a mean prediction-based reasoning approach for outcome prediction calculation. There were various numbers of segments in each audio file. To determine the correctness of the entire audio recording, the accuracy of each segment was calculated. If the model correctly predicted more than 50% of the audio files, the accuracy was deemed acceptable. Figure 10 presents an example of one speech file segmented into four segments (S1, S2, S3, and S4). Each segment is used as an input for the proposed model and the outcome for each segment is shown in Figure 10. In the final step, the average of all segments for one file is calculated. The maximum average is the predicted emotion.

As an optimizer, the Adam adaptive gradient descent algorithm [47] was utilized for training and the learning rate was set at 0.001. A softmax layer was applied following training to achieve a prediction accuracy output for various emotions or speakers. The training procedure was executed ten times for a maximum of 200 epochs, with 128 samples per batch, using Google Colab Pro [48] and the following settings: hardware accelerator: GPU; runtime structure: High-RAM. This model was developed using TensorFlow [49] and Keras [50] as the front-end system.

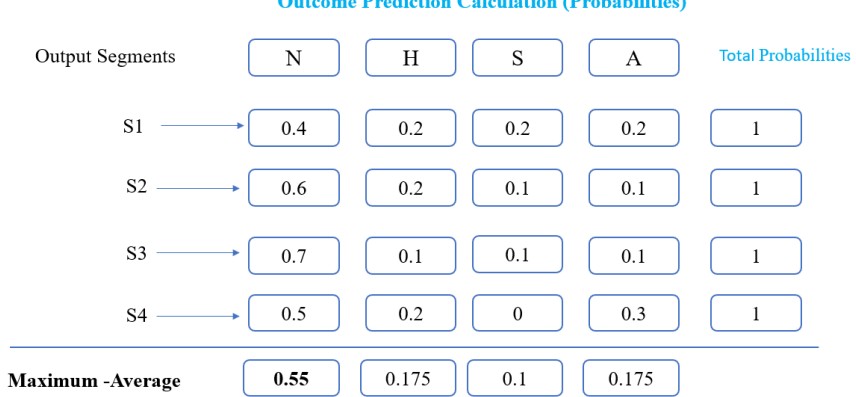

**Figure 10.** Outcome prediction calculation.

### 5.2.4. Discrete Emotion Recognition Results

The average accuracy of 10 runs, the standard deviation, and the highest and lowest accuracy are illustrated in Table 9 for each set separately (females, males, and females with males) and for the data in the four original, converted, merged, and crossed cases. Figure 11 shows that, generally, males perform better than females in all data types, where the average male accuracy reached 77.34% compared with 71.41% for converted females. It is evident that the categorization accuracy for the crossed data is the lowest compared with other data categories. As indicated previously, the male result is likewise the highest, at 59.16%, compared with 55.59% for females and 52.75% for all data, males and females combined.

**Table 9.** Emotion recognition average accuracy (%) results of 10 runs (Ori—original, Conv—converted, Mer—merged, Cro—crossed).

| | Emotions | | | | | | | | | | | |
|---|---|---|---|---|---|---|---|---|---|---|---|---|
| | **Female** | | | | **Male** | | | | **MF** | | | |
| | **Ori** | **Conv** | **Mer** | **Cro** | **Ori** | **Conv** | **Mer** | **Cro** | **Ori** | **Conv** | **Mer** | **Cro** |
| **Average** | 80.38 | 71.41 | 79.44 | 55.59 | 85.48 | 77.34 | 82.41 | 59.16 | 82.52 | 66.17 | 78.78 | 52.75 |
| **Std** | 4.50 | 3.21 | 4.50 | 6.74 | 3.73 | 5.57 | 3.40 | 3.06 | 4.83 | 3.72 | 3.10 | 1.12 |
| **Highest** | 86.36 | 76.56 | 84.18 | 65.94 | 90.37 | 87.50 | 86.93 | 62.50 | 88.72 | 71.09 | 82.74 | 54.09 |
| **Lowest** | 71.21 | 67.19 | 71.43 | 45.00 | 79.26 | 68.75 | 75.83 | 52.50 | 74.06 | 57.81 | 75.13 | 50.79 |

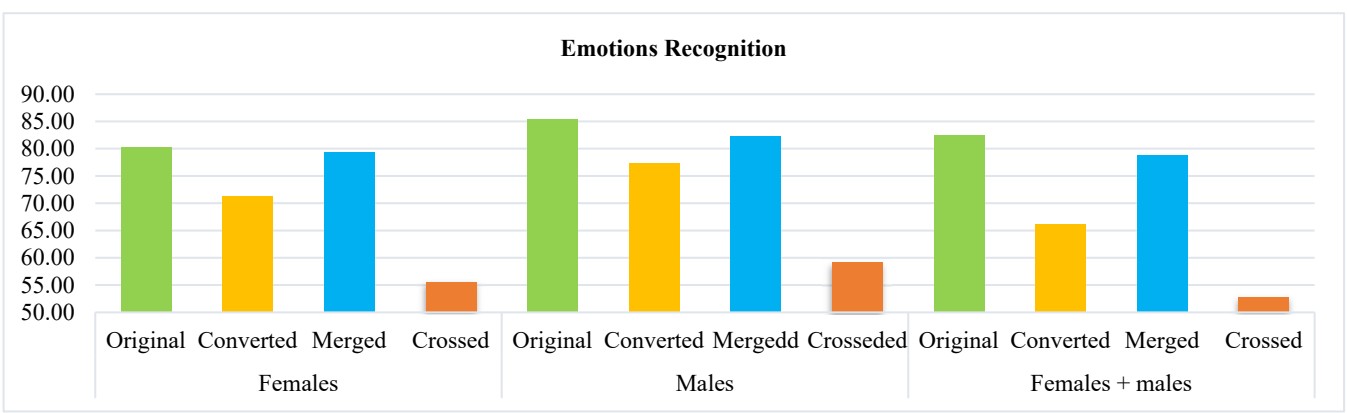

**Figure 11.** Emotion recognition results.

Regarding the merged data, the accuracy percentage in the merged data declined by 0.94% points for females and by 3.74% points for males compared with the original data. The difference in the classification accuracy percentage between the original and the converted data sets presents more in-depth comparison. Although the classification

accuracy score for males is generally higher, the standard deviation scale of the male group is larger than that of the female group. In addition, the gap between the highest and lowest classification accuracy for the female group is 9.37%, whereas for the male group it is 18.75%. Additionally, compared with males, the accuracy of females decreased less in the merged data compared with the original data.

The confusion matrices that were obtained after running 10 iterations using the original database, converted data, merged data, and crossing data are presented in Table 10.

**Table 10.** The average 10 runs confusion matrices of original, converted, merged, and crossed data for females, males, and males and females.

| | | Female | | | | Male | | | | MF | | | |
|---|---|---|---|---|---|---|---|---|---|---|---|---|---|
| | | **N** | **H** | **S** | **A** | **N** | **H** | **S** | **A** | **N** | **H** | **S** | **A** |
| **Original** | **N** | 78.2 | 9.2 | 8.9 | 3.8 | 84.1 | 10 | 4 | 2 | 81.6 | 9 | 5.8 | 3.6 |
| | **H** | 8.5 | 75.1 | 0.8 | 15.6 | 9.4 | 80.6 | 1.5 | 8.5 | 8.1 | 80.5 | 0.5 | 10.8 |
| | **S** | 13.2 | 2.9 | 80.2 | 3.7 | 12.1 | 1.8 | 85.6 | 0.5 | 18.3 | 1.9 | 77.8 | 1.9 |
| | **A** | 1 | 9.6 | 0.6 | 88.8 | 0.3 | 7.3 | 0.6 | 91.7 | 1 | 6 | 0.7 | 92.3 |
| | **Acc** | **80.38 ± 4.50** | | | | **85.48 ± 3.73** | | | | **82.52 ± 4.83** | | | |
| **Converted** | **N** | 65.4 | 3.1 | 24.6 | 6.9 | 71.2 | 16.4 | 12.4 | 0 | 63.1 | 16.1 | 14.3 | 6.5 |
| | **H** | 4.7 | 76.8 | 0 | 18.4 | 19.1 | 72.4 | 5.3 | 3.3 | 19.8 | 56.8 | 4.5 | 18.9 |
| | **S** | 31.7 | 0.5 | 67.2 | 0.5 | 7.1 | 8.6 | 83.6 | 0.7 | 13.4 | 6.4 | 75.3 | 4.8 |
| | **A** | 6.7 | 17.9 | 0 | 75.4 | 1.2 | 14.6 | 1.2 | 83 | 9.8 | 17.6 | 4.4 | 68.1 |
| | **Acc** | **71.41 ± 3.21** | | | | **77.34 ± 5.57** | | | | **66.17 ± 3.72** | | | |
| **Merged** | **N** | 74 | 9.3 | 12 | 4.7 | 78.3 | 12.4 | 8.3 | 1 | 75.5 | 11.8 | 8 | 4.8 |
| | **H** | 5.7 | 77.3 | 0.4 | 16.6 | 9.1 | 78.9 | 2.8 | 9.1 | 12.5 | 74.2 | 1.1 | 12.2 |
| | **S** | 14 | 3 | 80 | 3 | 9.1 | 3.9 | 84.2 | 2.8 | 13.5 | 3.4 | 78.4 | 4.7 |
| | **A** | 2.8 | 9.7 | 0.5 | 87.1 | 1 | 9.4 | 1.8 | 87.8 | 3.3 | 8.1 | 1.1 | 87.5 |
| | **Acc** | **79.44 ± 4.50** | | | | **82.36 ± 3.50** | | | | **78.78 ± 3.10** | | | |
| **Crossed** | **N** | 55.5 | 10.9 | 16.7 | 16.9 | 72.7 | 17.8 | 9.4 | 0.2 | 49.8 | 23.1 | 17.1 | 10 |
| | **H** | 8.7 | 50.2 | 7.9 | 33.2 | 14.7 | 43.9 | 5.3 | 36 | 19.8 | 41.3 | 9 | 30 |
| | **S** | 25.2 | 4.3 | 67.2 | 3.4 | 18.4 | 16 | 55.9 | 9.7 | 16.7 | 9.6 | 56.8 | 16.9 |
| | **A** | 6.8 | 14.6 | 3.2 | 75.4 | 5.3 | 15.9 | 1.7 | 77.2 | 8.2 | 9.7 | 3.4 | 78.7 |
| | **Acc** | **58.16 ± 4.34** | | | | **57.84 ± 6.06** | | | | **50.68 ± 3.15** | | | |

Figure 12 displays each emotion independently for the four cases (for females, males, and females and males). In the case of the original, merged, and crossed data, the emotion of anger obtained the highest accuracy in the case of the females, males, and both females and males. However, in the case of the converted data, the emotion of happiness obtained the highest accuracy in the females, and the emotion of sadness got the highest accuracy in the case of the males, and both females and males. When compared with the emotions of anger and sadness, the classification accuracy for the emotions of happiness and neutrality was the lowest.

The accuracy of sadness corresponds with the results of the sadness that were discussed above, where the sadness conversion results, in general, are good.

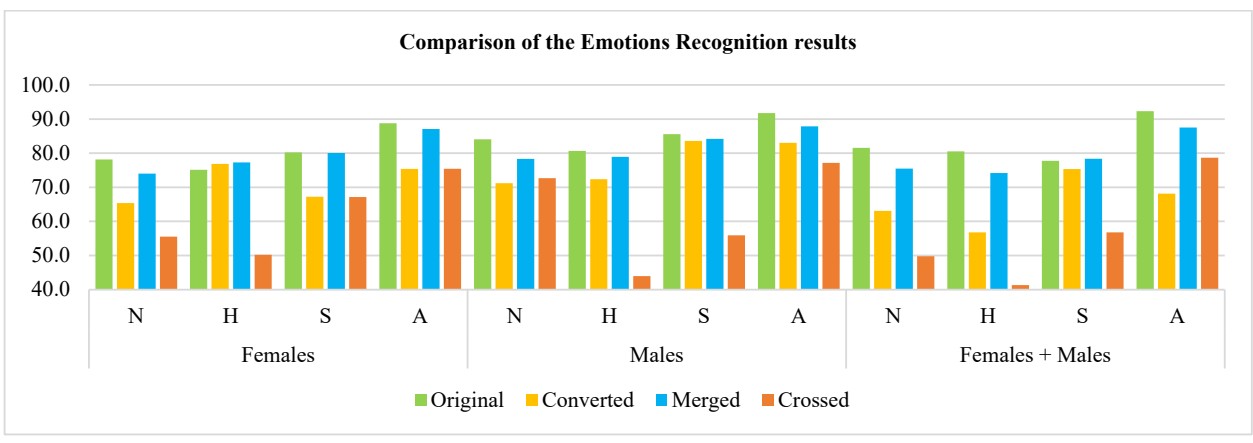

**Figure 12.** Emotion classification comparison.

5.2.5. Arousal–Valence Emotion Recognition Results

Arousal and valence are the most commonly two-dimensional spaces used to understand emotions according to what occurs within a dimensional space. Emotional arousal relates to the intensity of the emotional state, which can be high or low, whereas emotional valence denotes the degree to which an emotion is positive or negative [51].

The accuracy of arousal–valence emotion recognition is shown in Table 11 for 10 runs of the original, converted, and merged data. We observe here that the accuracy of the converted data due to arousal (high and low) is equivalent to the accuracy of the original data and might even grow significantly in the case of females. In the case of the merged data, the overall accuracy improved significantly. In contrast, the accuracy of the converted data for the valence emotion was generally lower than that of the original data. Regarding the accuracy of the results with regard to gender, we find that the results of valence are comparable to the previous results in general, as the accuracy for males is higher than that of females, except in the case of the merged data. In contrast to the emotion of arousal, females had a higher rate of accuracy.

**Table 11.** Average accuracy of 10 runs for arousal–valence emotion recognition.

|  |  | Original | | | Converted | | | Merged | | |
|---|---|---|---|---|---|---|---|---|---|---|
|  |  | **Female** | **Male** | **MF** | **Female** | **Male** | **MF** | **Female** | **Male** | **MF** |
| **Arousal** | **Acc** | 92.27 | 81.09 | 82.73 | 95.47 | 81.09 | 82.73 | 95.39 | 87.89 | 87.73 |
|  | **STD** | 0.02 | 0.07 | 0.04 | 0.02 | 0.07 | 0.04 | 0.01 | 0.03 | 0.03 |
| **Valence** | **Acc** | 85.15 | 87.56 | 88.08 | 72.34 | 81.72 | 80.63 | 87.34 | 84.06 | 88.13 |
|  | **STD** | 0.04 | 0.04 | 0.02 | 0.06 | 0.06 | 0.04 | 0.05 | 0.03 | 0.03 |

5.2.6. Speaker Identification Results

Table 12 provides a summary of the results of speaker identification using the original, converted, merged, and crossed database, which includes the mean accuracy for 10 runs, higher and lower accuracy, and the standard deviation. The results are presented for males, females, and males and females together. For further understanding, Figure 13 presents the overall average accuracy. According to the table and the figure, the identification accuracy of female speakers was 94.97% greater than that of male speakers, which was 91.85%, and greater than male and female speakers combined, which was 91.44%. Compared with the converted data, we noticed that the accuracy of identifying male and female speakers is comparable, with a small increase for males. The accuracy of speaker identification reduced somewhat (0.5% for males, 2.66% for females, and 3.07% for all speakers) when the original data were combined with the converted data. In the case of crossed data, males have the highest accuracy, as depicted in the image.

**Table 12.** Summary of results of 10 runs of speaker recognition (Ori—original, Conv—converted, Mer—merged, Cro—crossed).

|  | Speakers | | | | | | | | | | | |
|---|---|---|---|---|---|---|---|---|---|---|---|---|
|  | Female | | | | Male | | | | MF | | | |
|  | **Ori** | **Conv** | **Mer** | **Cro** | **Ori** | **Conv** | **Mer** | **Cro** | **Ori** | **Conv** | **Mer** | **Cro** |
| **Average** | 94.97 | 85.62 | 92.31 | 50.19 | 91.85 | 86.88 | 91.34 | 66.28 | 91.43 | 83.44 | 88.36 | 56.45 |
| **Std** | 3.45 | 5.30 | 1.52 | 4.13 | 3.10 | 4.11 | 2.76 | 5.02 | 4.73 | 4.60 | 3.64 | 5.17 |
| **Best** | 98.86 | 93.75 | 94.54 | 56.11 | 95.83 | 95.31 | 96.12 | 76.03 | 96.78 | 88.28 | 93.62 | 64.94 |
| **Worst** | 87.43 | 76.56 | 89.50 | 44.83 | 86.31 | 81.25 | 87.93 | 59.94 | 83.04 | 75.78 | 83.19 | 49.53 |

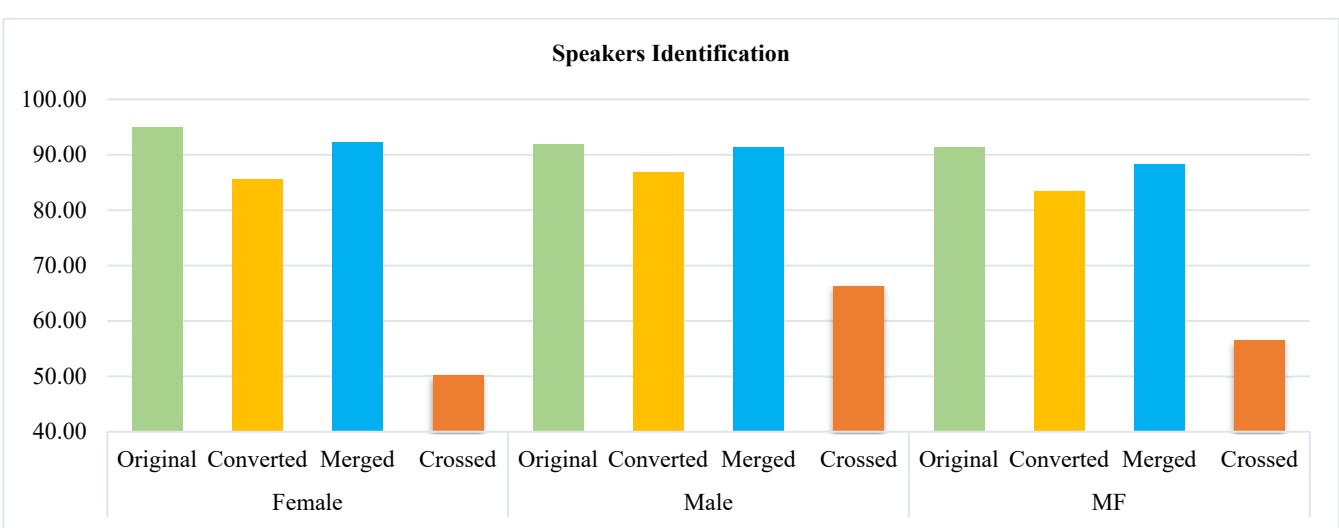

**Figure 13.** Speaker identification results.

Comparing the values of the standard deviation and the difference between the highest and lowest accuracy, the value of the standard deviation for females in both the original data and the converted data is higher than the value for males, but in the case of the merged data it is the lowest.

In the summary of the evaluation results in general, whether using the spectrum and prosody conversion or automatic recognition, we found that the results of males are, in general, better than the results of females, with a few notable exceptions. This finding is consistent with the outcomes of our previous study in emotion recognition regarding the whole of the database, whether such investigations were carried out with the human perceptual test or with the CRNN classifier [44,45]. Regarding the used pre-trained model, the overall accuracy of emotion recognition utilizing translated data ranged from 66.17% to 77.33%, which we consider to be an acceptable level of accuracy when utilizing the pre-trained model.

## 6. Conclusions

The large gap in the research related to voice conversion in general between Arabic and other languages such as English and Chinese motivated our study. Our purpose is to answer whether it is possible to convert Arabic voice emotion using a model that has already been trained. This study conducted emotional voice conversion using StarGANv2-VC, i.e., StarGANv2-VC-Arabic Emotional, where the F0 and ASR models were trained in the English language. The evaluation process of the emotional voice conversion was performed through a prosody and spectrum conversion in addition to the state-of-the-art deep learning classifier CRNN model. The results show that StarGANv2-VC, with a pre-trained model in English, can be used to perform an Arabic emotional voice conversion mainly through spectral envelope conversion. The evaluation results revealed that male voices were rated higher than female voices and that the conversion from neutral to other emotions received a higher evaluation score than the conversion of other emotions. However, a different tendency was observed in the evaluation results depending on the source and target emotional states. The experiment's performance is promising when compared with state-of-the-art emotional voice conversion techniques. We believe that this finding is a good starting point but that it can be improved and that better results are possible. Ongoing work consists of taking advantage of the obtained results to improve the EVC of the Arabic language by integrating the F0 and ASR models that are completely trained in the Arabic language.

**Author Contributions:** Conceptualization, A.H.M., Y.A.A. and S.-A.S.; methodology A.H.M., Y.A.A. and S.-A.S.; software, A.H.M.; validation, A.H.M.; formal analysis, Y.A.A.; investigation, S.-A.S.; resources, A.H.M.; data curation, A.H.M.; writing—original draft preparation, A.H.M.; writing—review and editing, Y.A.A. and S.-A.S.; visualization, A.H.M.; supervision, Y.A.A. and S.-A.S.; project administration, A.H.M.; funding acquisition, Y.A.A. All authors have read and agreed to the published version of the manuscript.

**Funding:** This work was partially supported by the Researchers Supporting Project number (RSP-2021/322), King Saud University, Riyadh, Saudi Arabia.

**Institutional Review Board Statement:** Not applicable.

**Informed Consent Statement:** Not applicable.

**Data Availability Statement:** Not applicable.

**Conflicts of Interest:** The authors declare no conflict of interest.

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
