# Peer review of "Arabic Emotional Voice Conversion Using English Pre-Trained StarGANv2-VC-Based Model"

_applsci, doi:10.3390/app122312159_

Round 1
Reviewer 1 Report
The paper is interesting, an experimental design and evaluation were performed, the results are promising and give insights for future works
The following are main suggestions for further improvements, others are left in the attached file.
- revise the paper and references, format and style
- to add main results in the abstract
- remove period before citation ; .[ ].
- cite this claim "...and autoencoder are the most common techniques used in EVC frameworks."
- define "ASR and F0" at first use in introduction section
- check https://ieeexplore.ieee.org/document/7073218 for preliminary work
- figure1: check caption, starGANv2-VC framework
- revise the tables/figures captions to be more expressive
- revise tables' presentation for better readability
- the dataset has 5 emotions class, while your exp using 4, why?
- "Due to the significant effort and resources required for subjective evaluation" ... to mention/describe this approach details at least , even if not considered, so reader can understand the challenge
- Figure 9 not cited/ref-ed

Reviewer 2 Report
The paper discusses Arabic emotions and their conversion to another form using a model trained for another language. A convolutional recurrent neural network (CRNN) is proposed for this purpose. The following points need to be considered:
· The summary of the results needs to be added to the abstract.
· It is not clear to me how you train the model. Also, did you use the Arabic language for training because the title is a little bit confusing?
· Figures 2 and 3 are not that clear—for example, what 20 represents and the colors as well in Figure 2. Also, the histograms in Figure 3 need an explanation.
· How does the following statement get classified, and how?
?assaadaat bat?alul ħarbi wassalaam
· Table 5 and Figures 5, 6, 7, and 8 need more elaboration.
· The results seem not seem that good. Is there a reason for that?
· Summarize the results in the conclusion section.
· Overall, the work is okay, but the description of the technical work may need more effort.
Round 2
Reviewer 2 Report
I do not have any further comment.